# Total Knee Arthroplasty for the Oldest Old

**DOI:** 10.3390/geriatrics6030075

**Published:** 2021-08-04

**Authors:** Carmen da Casa, Helena Fidalgo, Javier Nieto, Enrique Cano-Lallave, Juan F. Blanco

**Affiliations:** 1Instituto de Investigación Biomédica de Salamanca (IBSAL), 37007 Salamanca, Spain; cdacasap@saludcastillayleon.es (C.d.C.); hfidalgogomez@gmail.com (H.F.); 2Rehabilitation Department, University Hospital of Salamanca, 37007 Salamanca, Spain; javiernieto@saludcastillayleon.es (J.N.); ecanol@saludcastillayleon.es (E.C.-L.); 3Orthopaedic Surgery and Traumatology Department, University Hospital of Salamanca, 37007 Salamanca, Spain

**Keywords:** total knee arthroplasty, octogenarians, septuagenarians, osteoarthritis, rehabilitation, physical therapy, older

## Abstract

The present study describes and compares the early functional results after total knee arthroplasty (TKA) of the oldest-old population (aged over 84 years) and a randomly matched younger septuagenarian cohort so treated. We aimed to evaluate the early functional outcomes after patients’ rehabilitation and the yearly requirements for hospital readmission and emergency room visits after TKA. We noted a similar length of hospital stay for octogenarian and septuagenarian patients, and we determined that both groups of patients were improving ROM (both flexion and extension) after the rehabilitation program (*p* < 0.05, in all cases), but there were no significant differences between octogenarian and septuagenarian improvement of the knee function (*p* > 0.05, in all cases). Patients from both age groups behaved similarly in terms of mobility before starting rehabilitation and after completion of the rehabilitation program. We noted that older octogenarian patients showed a higher one-year hospital readmission rate than younger septuagenarian patients, but similar early emergency room visits for both age groups. The findings of this study allow us to conclude that advanced age in itself should not be a contraindication for TKA.

## 1. Introduction

Increased life expectancy and improved healthcare have led to an increasing number of older patients requiring an arthroplasty procedure [1,2]. The indication would range from the treatment of fractures to osteoarthritis or the treatment of other articular diseases. Advanced age should not be in itself a contraindication for arthroplasty [3,4]. A significant number of older patients lead an active life and may therefore require this type of surgical treatment [5].

Total knee arthroplasty (TKA) relieves or eliminates joint pain caused by osteoarthritis and improves joint function. It presents very satisfactory results, with low complication rates and long implant survival [6,7]. Advanced age can be a determining factor in the assessment of the clinical and functional status of patients, bearing in mind the necessary measures to reduce the surgical risk [6,8]. Likewise, the bone tissue of older patients may condition the type of implant, seeking stabilization and immediate anchorage to allow the early resumption of activity. Currently, cemented implants are widely accepted for use in older patients [9], but also other authors have reported satisfactory results of non-cemented implants in older patients [10].

Advanced age is often associated with a higher rate of comorbidity, which could hinder the surgical procedure and cloud the prognosis. Clinical optimization and control of associated pathologies are essential for a satisfactory outcome [4]. Despite it, rehabilitation after a total hip replacement has been reported to be a common practice [11]. There is a consensus on the benefit of rehabilitation in the process of TKA [12]. Several protocols and techniques are used without evidence of the superiority of one over the other [12,13,14]. Rehabilitation after TKA includes physical exercises and other modalities (like hydrotherapy or electrotherapy, among others) in order to reduce the pain and improve joint function [11,13,14]. Another topic to consider in rehabilitation, in older patients, is the need for collaboration in this type of treatment. The ability of the patients to participate in these therapies could be influenced by some factors, like clinical situations, age, or cognitive impairments [15].

The evaluation of the results of TKA could be performed by diverse tools such as quality of life questionnaires, complication rates estimation, or joint function assessment. One of the related parameters to the outcome after TKA is the knee function evaluation by the range of motion (ROM) assessment and measuring the improvement of the patient’s mobility [16,17].

The purpose of the present study is to describe and compare the early functional results after TKA of the oldest-old population (aged over 84 years) and a randomly matched younger cohort so treated. We aimed to evaluate the early functional outcomes after patients’ rehabilitation and the yearly requirements for hospital readmission and emergency room visits after TKA.

## 2. Materials and Methods

We have conducted a retrospective study including all patients over 84 years of age surgically treated by TKA on a single tertiary teaching hospital, between 1 January 2016 and 16 March 2020. A balanced 1:1 random sample of patients aged 71–75 from the same study period was used for results comparisons. The study population comprised 52 octogenarian patients undergoing TKA and 52 septuagenarian TKA patients.

Data collection from patient’s medical records included demographic variables such as age, gender, diagnosis, and laterality. We have also collected data regarding the rehabilitation program length, and patient’s ROMs, as well as the Physical Red Cross Scale (PRCS) score, once the surgery was completed and before starting physical therapy and once rehabilitation was completed. The PRCS evaluates the physical ambulatory ability of the patient. It is a tool widely used in Spain through the Comprehensive Geriatric Assessment (CGA). The physical-status evaluation of PRCS concerns 5 levels of ambulatory ability from 0, which indicates full capability, to 5, which indicates no ambulatory capability [18].

After hospital discharge, all patients underwent a one-year follow-up, noting whether the patient needed hospital readmission or an emergency room visit.

### Statistical Analysis

Statistical analysis was performed using the IBM^®^ SPSS^®^ Statistics program (v.26). Descriptive statistics included mean, standard deviation, and range. The normality of sample distribution was defined by the Kolmogorov-Smirnoff test (Lilliefors corrected), showing that all variables are distributed according to a Gaussian distribution. We ascertain the statistically significant differences among groups by Student’s *t*-test for quantitative variables and Chi-Squared tests for qualitative variables. To compare the groups before and after rehabilitation, the Student’s *t*-test for paired samples was used on quantitative variables, and Cochran’s Q test was used for paired categorical variables. In all cases, *p* ≤ 0.05 was considered statistically significant.

## 3. Results

### 3.1. Demographic Characteristics

During the five-year study period, a total of 52 patients aged over 84 years underwent a TKA at our center. A balanced study was conducted with younger septuagenarian TKA patients aged from 71 to 75, acting as a control group. The demographic characteristics of each group are presented in Table 1. There were no statistical differences (*p* > 0.05), despite the age classification established.

The mean length of stay (LOS) was 6.10 ± 1.68 days for octogenarian patients, and 5.77 ± 1.57 for septuagenarian patients (*p* > 0.05).

### 3.2. Rehabilitation Outcome

A total of 45 older octogenarian patients (86.5%) started the rehabilitation program during the initial hospital admission, reaching 90.4% after a maximum of six days after hospital discharge. There were five (9.6%) octogenarian patients from whom we could not record rehabilitation outcomes, as four of them underwent private rehabilitation in different nursing homes, and one patient suffered a surgical wound infection requiring a new intervention. Regarding younger septuagenarian patients, 94.2% started rehabilitation during the initial hospital admission, reaching 96.2% after a maximum of four days after hospital discharge. Two (3.8%) patients refused to attend any rehabilitation program.

The baseline flexion and extension ROM before and after the physical therapy of the rehabilitation program are noted in Table 2.

We determined that both groups of patients were improving in ROM (both flexion and extension) after the rehabilitation program (*p* < 0.05, in all cases), but there were no significant differences between octogenarian and septuagenarian improvement of the knee function (*p* > 0.05, in all cases). Patients from both age groups behaved similarly before starting rehabilitation and after completion of the rehabilitation program.

Therefore, we compared the PRCS before and after rehabilitation in both age groups (Table 3). We noted that both groups of patients were improving in PRCS after the rehabilitation program (*p* < 0.05, in all cases), and again there were no significant differences between octogenarian and septuagenarian patients’ improvement (*p* > 0.05, in all cases).

Older octogenarian patients received a mean of 10.9 ± 7.5 rehabilitation sessions, while younger septuagenarian patients, a mean of 15.8 ± 13.6 rehabilitation sessions (*p* > 0.05).

### 3.3. Hospital Readmissions and Emergency Room Visits

The follow-up for every patient comprised one year after hospital discharge. We only noted one patient dying during the second month of follow-up in the older octogenarian group.

Regarding emergency room (ER) visits, 46.2% of older octogenarian patients and 28.8% of younger septuagenarian patients visited the emergency department the following year after hospital discharge (*p* > 0.05). Regarding emergency department visits during the first 30-day follow-up after discharge, 24.0% of older octogenarian patients and 25.0% of younger septuagenarian patients visited the emergency department (*p* > 0.05), and 56.0% of octogenarian patients and 43.8% of septuagenarian patients visited the emergency department within the first 90 days of follow-up (*p* > 0.05). The mean time to emergency room visits of octogenarian patients was 99.5 ± 90.3 days, and 132.8 ± 107.7 days for septuagenarian patients (*p* > 0.05).

A total of ten (19.2%) older octogenarian patients required hospital readmission during the following year after TKA (one of them experienced two hospital readmission episodes). On the other hand, two (3.8%) younger septuagenarian patients experienced a single hospital readmission during the follow-up period (*p* = 0.014). The readmission episodes were carried out in different hospital departments, resumed in Table 4. Orthopedic Surgery and Traumatology department received 3 (33.3%) of octogenarian readmission cases and 0% of septuagenarian cases. The mean time to readmission of octogenarian patients was 149.4 ± 90.5 days, and 330.0 ± 25.5 days for septuagenarian patients (*p* = 0.022).

Attending to the early readmissions (up to 90 days after initial hospital discharge), in the octogenarian patients’ group, there was one patient who was readmitted to the Orthopedic Surgery and Traumatology Department within the first 30 days after hospital discharge (the one who required a re-intervention due to the surgical wound infection), and three (5.8%) octogenarian patients were readmitted within the first 90 days of follow-up after initial hospital discharge (one of them was also readmitted to the Orthopedic Surgery and Traumatology department). There were no early readmissions on the septuagenarian group.

## 4. Discussion

The main finding of this work was to report that TKA could offer comparable results in terms of knee function and mobility improvement in younger septuagenarian and the oldest octogenarian patients, with a novel insight regarding patient readmission and emergency room visits. The possible relationship of age with the results of TKA has been the subject of several studies that showed that no age limit could be placed on the performance of this surgical procedure [2,4,19,20]. Our results agree with previous works by other authors [2,21], which report comparable results among patients of different age groups, showing that advanced age is not a limitation *per se* for the performance of TKA.

Previous studies on TKA for older patients pointed to a longer hospital LOS for older patients [2,22]. However, we observed no statistically significant differences between oldest-old patients and septuagenarians in terms of LOS [16]. Despite it, our results agree with previously pointed satisfactory results of TKA in older patients in terms of analgesics and improved function reported by Maempel et al. [22].

It is accepted that rehabilitation programs, also based on physiotherapy, improve the outcome of TKA patients. Diverse protocols and techniques have been associated with improvements in function and patient perception after total knee arthroplasty [11,12,13,14]. There is no agreement in the scientific literature on which type of physiotherapy is most effective in patients undergoing total knee arthroplasty. Protocols for preoperative initiation of physical therapy have been proposed that may improve outcomes. Papalia et al. [14] suggest that patients over 65 years of age would benefit from intensive physiotherapy and aqua therapy protocols in conjunction with physical activity.

In our study population, most patients initiated the rehabilitation treatment during the initial hospital admission and continued after hospital discharge. No age-related difficulties in carrying out rehabilitation treatment were found, except for one case that became infected and required further surgical treatment. When we analyzed the effect of early rehabilitation in both groups, we found that this treatment significantly improved the functional situation of the patients in both the ROM and the PRCS. We found no significant differences between groups when we analyzed the functional improvement of the patient; these findings agree with the results reported by Kuperman et al. [16] in which they also found no differences between patients groups. Our findings confirm that at least in the group of oldest octogenarian patients studied, age is not a limitation for conducting the postoperative rehabilitation treatment.

Despite the functional outcome of older TKA patients, some studies pointed to a worse quality of life, medical complications incidence, and higher mortality for patients over 80 years [2,19]. In this regard, we noted that older octogenarian patients showed a higher one-year hospital readmission rate than younger septuagenarian patients. However, we should bear in mind that most patients attended non-orthopaedical hospital departments, and hence these readmissions were hardly related to the TKA procedure. On the other hand, we noted similar early ER visits for both age groups, which would suggest that hospital readmissions may be explained by higher comorbidity for the oldest-old patients, which although is not the subject of the present study, has been also previously reported [16].

This fact leads us to quote the limitations of the present study, mainly due to the sample size and the lack of data on patient comorbidity. Our focus on the analysis of knee function and patients’ mobility provides a global view of functional outcomes after TKA on the oldest-old patients.

## 5. Conclusions

The findings of this study allow us to conclude that advanced age should not be in itself a contraindication for TKA.

When considering knee function in ROM terms and patients’ mobility after TKA, similar results are to be expected for the oldest-old patients. Older age has been associated with a higher one-year readmission rate, and further analysis would be necessary to prove it as an expression of higher comorbidity.

## Figures and Tables

**Table 1 geriatrics-06-00075-t001:** Demographic characteristics of each study group.

Variables		Older Octogenarians	Younger Septuagenarians	*p*-Value
Age (years)	Mean ± SD [Range]	84.9 ± 1.2 [84–89]	73.3 ± 1.5 [71–75]	<0.001
Gender	Women (%)	61.5%	57.7%	0.689
Diagnosis	Primary osteoarthritis	96.2%	84.6%	0.541
	Secondary osteoarthritis	3.8%	13.4%	
Laterality *	Right (%)	55.8%	50.0%	0.347

SD: Standard deviation. * The laterality value refers to the episode analyzed in the present study, avoiding duplication of values. In the group of younger septuagenarians, there were seven patients (13.46%) who presented a new contralateral TKA episode in the five years covered by the study. Two patients in the older octogenarian group (3.85%) had previously undergone a contralateral TKA procedure during the study period.

**Table 2 geriatrics-06-00075-t002:** Range of motion measurements before and after rehabilitation: knee flexion and extension.

ROM	Octogenarians (*n* = 47)	Septuagenarians (*n* = 50)	*p*-Value (between Groups)
Flexion before physical therapy	61.12⁰ ± 18.09⁰ [30⁰–100⁰]	64.22⁰ ± 17.07⁰ [30⁰–95⁰]	0.381
Flexion after physical therapy	100.13⁰ ± 8.23⁰ [85⁰–120⁰]	99.38⁰ ± 9.08⁰ [70⁰–115⁰]	0.689
Difference in flexion between before and after	35.26⁰ ± 18.42⁰ [5⁰–70⁰]	34.17⁰ ± 19.11⁰ [0⁰–85⁰]	0.789
***p*-value (within groups)**	<0.001	<0.001	
Extension before physical therapy	−6.43⁰ ± 9.30⁰ [−40⁰–0⁰]	−6.57⁰ ± 9.08⁰ [−40⁰–0⁰]	0.939
Extension after physical therapy	−3.46⁰ ± 5.15⁰ [−20⁰–0⁰]	−3.33⁰ ± 6.55⁰ [−30⁰–0⁰]	0.921
Difference in extension between before and after	3.82⁰ ± 8.58⁰ [−15⁰–30⁰]	3.54⁰ ± 6.01⁰ [−5⁰–20⁰]	0.862
***p*-value (within groups)**	0.009	<0.001	

**Table 3 geriatrics-06-00075-t003:** **Physical Red Cross Scale** (PRCS) scores before and after rehabilitation.

PRCS	Octogenarians (*n* = 47)	Septuagenarians (*n* = 50)	*p*-Value (between Groups)
Before physical therapy	3 (100%)	3 (98.0%) 2 (2.0%)	0.734
After physical therapy	3 (10.5%) 2 (68.4%) 1 (21.1%)	3 (18.8%) 2 (60.4%) 1 (20.8%)	0.802
Difference between before and after	Equal (10.5%) −1 (68.4%) −2 (21.1%)	Equal (20.8%) −1 (58.4%) −2 (20.8%)	0.421
***p*-value** **(within groups)**	<0.001	<0.001	

**Table 4 geriatrics-06-00075-t004:** Hospital readmission departments and causes on the following year after TKA.

	Angiology Vascular Surgery	Cardiology	General Gastrointestinal Surgery	Digestive	Neurology	Orthopaedic Surgery
Septuagenarians (71–75 years)	Chronic arterial ischemia		Hemorrhoid (Grade IV)			
Octogenarians (≥84 years)		Sinus dysfunction Peripheral arterial disease	Acute cholecystitis Inguinal hernia	Zenker’s diverticulum Cholangitis	Stroke	Prosthesis infection Pain and functional impotence Diaphyseal humeral fracture

## Data Availability

All datasets generated in the study are available from the corresponding author upon reasonable request.

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
