# Peer review of "Total Knee Arthroplasty for the Oldest Old"

_geriatrics, 2021, doi:10.3390/geriatrics6030075_

Round 1

Reviewer 1 Report

  1. Line 93 : 't-Student tests" should be "Student's t-test"
  2. Line 94 : " Chi-Square" should be "Chi-Squared"
  3. Line 95 : "Student's-T test" should be "Student's t-test"
  4. all the "p<=0.05", the letter "p" should be in italian style "p"
  5. Line 115 : "Rehabilitation outcome." should be "Rehabilitation outcome", no period sign
  6. Line 128 : "improving ROM" should be "improving in ROM"
  7. Line 134 : "improving PRCS" should be "improving in PRCS"
  8. Line 144 " We only noted one...", should describe more about how and why that one patient died.
  9. Line 149 : should explain/express more clear about the phrase "Attending to the shorter term, 24.0%............"
  10. Line 158 : "process" could be removed
  11. Line 201 : "development of postoperative..." maybe better changed to "conducting the postoperative... 
  12. Line 203 : "Despite de functional outcome" should changed to english style 
  13. Line 205 : "over 80 years [2,20] .In..." should be "over 80 years [2,20]. In..."
  14. Line 206 : should explain more about why the author jump into this discussion conclusion "However, we should bear in mind that most patients attended non-orthopaedical hospital departments, and hence these readmisions were hardly related to the TKR procedure."

Reviewer 2 Report

Dear authors,

I appreciated the paper by da Casa and colleagues on the use of TKA in older patients. Ageing of the population is increasing worldwide, and consequently, the request for TKA will increase in the next years. Finding new solutions and indications for this procedure is, therefore, necessary to manage the economic burden of this procedure.

The paper is well written and interesting. However, I have only some concerns.

Page 4 - line 149. authors reported that "Regarding emergency room (ER) visits, 46.2% of older octogenarian patients and 147 28.8% of younger septuagenarian patients visited the emergency department the following 148 years after hospital discharge (p>0.05)". However 28.8% and 46.2% are very different percentages, are the authors sure that there is no statistical significance?

Which were the causes of hospital readmissions? it was possible to match them with the comorbidities of the patients? (maybe using the charlson comorbidity index)

One big limitation of this study is the short follow up. One year is nothing for older patients. The economic burden of TKA is relevant, therefore, perform TKA also in very in old patients could be dangerous for a public healthcare system, because the life expectancy of these patients is usually not so long. The authors should discuss this point. 

Moreover, another limitation of this study is the lack of apriori power analysis. Are these patients enough to make relevant conclusions?

best regards

Round 2

Reviewer 2 Report

Ok